# Clay-Catalyzed Ozonation of Hydrotalcite-Extracted Lactic Acid Potential Application for Preventing Milk Fermentation Inhibition

**DOI:** 10.3390/molecules27196502

**Published:** 2022-10-01

**Authors:** Meriem El Baktaoui, Nour El Houda Hadj-Abdelkader, Amina Benghaffour, Vasilica-Alisa Arus, Nadia Bennani-Daouadji, Fatiha Belkhadem, René Roy, Abdelkrim Azzouz

**Affiliations:** 1Nanoqam, Department of Chemistry, University of Quebec at Montreal, Montreal, QC H3C3P8, Canada; 2LEMFN, Department of Materials Engineering, University of Science and Technology, El M’naouer, Oran BP-1505, Algeria; 3Catalysis and Microporous Materials Laboratory, Vasile Alecsandri University of Bacau, 600115 Bacau, Romania; 4École de Technologie Supérieure, Montreal, QC H3C 1K3, Canada

**Keywords:** lactic acid, ozonation, clay catalyst, hydrotalcite, montmorillonite

## Abstract

An unprecedented route for mitigating the inhibitory effect of lactic acid (LA) on milk fermentation was achieved through lactate adsorption on hydrotalcite (Ht) from simulated lactate extracts. During its regeneration by ozonation, Ht displayed catalytic activity that appeared to increase by addition of montmorillonite (Mt). Changes in the pH, Zeta potential and catalyst particle size during LA ozonation were found to strongly influence LA–LA, LA–catalyst and catalyst–catalyst interactions. The latter determine lactate protonation–deprotonation and clay dispersion in aqueous media. The activity of Mt appears to involve hydrophobic adsorption of non-dissociated LA molecules on silica-rich areas at low pH, and Lewis acid–base and electrostatic interactions at higher pH than the pKa. Hydrotalcite promotes both hydrophobic interaction and anion exchange. Hydrotalcite–smectite mixture was found to enhance clay dispersion and catalytic activity. This research allowed demonstrating that natural clay minerals can act both as adsorbents for LA extract from fermentation broths and as catalysts for adsorbent regeneration. The results obtained herein provide valuable and useful findings for envisaging seed-free milk clotting in dairy technologies.

## 1. Introduction

Lactic acid (LA) is a 2-hydroxypropionic acid with a wide variety of industrial and biotechnological applications [1,2,3,4,5,6]. LA has negative impacts on animal metabolisms [4] and on the growth of Lactic Acid Bacteria (LAB) in dairy technology being produced by anaerobic and O_2_-poor fermentation broths [7,8,9,10]. Milk clotting triggers via lactose fermentation that induces pH decrease, but excessively low pH favors lactate anion protonation into non-dissociated LA molecules. The latter can easily diffuse through the LAB cell membrane, thereby modifying the internal pH and cell metabolism [11,12]. A major challenge barely tackled so far resides in controlling LA concentration during milk clotting. More or less successful LA removal attempts have been reported in the literature [13,14,15,16,17,18,19,20,21,22,23,24,25,26,27], but cannot be applied to dairy technologies. Continuous LA removal during fermentation using food-compatible adsorbents is a very viable approach in this regard [28,29,30]. Thus, the present research was undertaken.

Layered double hydroxides (LDHs) are available anionic clay minerals with chemical stability, surface basicity and total lack of toxicity. LDH and hydrotalcite (Ht), their natural and lower-cost carbonate-exchanged counterpart, can act as catalysts and adsorbents for acidic species for diverse purposes including stomach hyperacidity treatment [19,29,31,32,33]. Ht can be used to remove lactate anion exchange from milk fermentation broth. Continuous or periodical regeneration in a separate carbonate-rich aqueous medium allows releasing lactate by reversed anion exchange. The use of ozone that converts LA into carbonate anion is expected to enhance Ht regeneration. This is a judicious strategy that has never been reported so far.

LA ozonation was already investigated in the absence [34,35] and presence of solid catalysts [36,37,38,39,40,41,42], low cost, non-toxic and widely available smectites such as bentonite and derivatives [36,37,38,39,40,41,42,43,44] but scarcely in the presence of hydrotalcite [34,35]. Among smectites, montmorillonite (Mt) can be used as co-catalyst to improve Ht regeneration by ozonation. Elucidation of a possible synergy is a major objective of the present work. Mt exhibits pH-dependent charges on silanols and lattice oxygen atoms that promote Lewis acid–base interactions (LABI) and hydrophilic character and induce higher dispersion in water as compared to hydrophobic LDH [31,45]. In this regard, the strength of LABI is undoubtedly the most important phenomena in aqueous media [37]. LABI may compete with electrostatic and organophilic-hydrophilic interactions which are also strongly pH-dependent. The resulting effect is expected to influence the adsorption of molecular LA or lactate anion according to the nature of the clay surface and the modification procedure [36,37,38,39,42,46,47,48,49,50].

LA is known to adsorb on basic surfaces such as LDH [29], but the contributions of the hydrophobic interactions of non-dissociated molecules and lactate anion exchange still remain to be assessed. The glaring lack of data related to LA interactions with clay catalysts in ozonation processes was the main reason for undertaking this research. Acid activation can influence the preponderance of a given interaction and catalytic activity due to decay in Al content, cation exchange capacity and surface charge density [37,51,52]. Measurements of the Zeta potential and clay particle size can be correlated to the catalytic activity and LA removal efficiency.

## 2. Experimental Section

### 2.1. Catalyst Preparation and Characterization

A commercial hydrotalcite (Fluka), denoted as Ht, was used as an anion exchanger for lactate extraction from dairy fermentation broth as described elsewhere [29,30]. In addition, highly crystalline Na^+^-exchanged montmorillonite (NaMt) was prepared through purification of a crude bentonite (Sigma-Aldrich, St. Louis, MO, USA) by removing volcanic ashes, dense silica phases and amorphous impurities (Appendix A). Thus, a mixture of 200 g of crude bentonite dispersed in 2 L of distilled water and 70 g of NaCl was vigorously stirred at 80 °C for 6 h, cooled overnight and centrifuged at 12,000 rpm. After supernatant removal, the resulting NaMt was prone to repeated washings in distilled water, followed by centrifugations and dialysis in cellulose bags immersed in vigorously stirred distilled water until total disappearance of chloride as monitored by AgNO_3_ tests and final drying overnight at 30–40 °C [39]. For comparison, KMt, CaMt and MgMt were prepared by stirring 3 g of NaMt with 1 g of metal salts (KCl, CaCl_2_ and MgCl_2_) in 100 mL of distilled water for 6 hours at 80 °C.

To investigate the effect of the exchangeable cation, additional Fe(II)Mt, Co(II)Mt, Ni(II)Mt and Cu(II)Mt samples were prepared according to the same procedure using corresponding metal chloride salts. The effect of the Si/Al ratio on the acid–base properties was studied using bentonite and acid-activated counterparts denoted as HMt-1, HMt-4, HMt-8, HMt-15 and HMt-24. Acid activation of the same raw bentonite was achieved with concentrated aqueous H_2_SO_4_ solution for different contact times (1, 4, 8, 15 and 24 h) according to a procedure described elsewhere [53]. X-ray diffraction analysis was performed using a D8 Advance Bruker instrument at a 40 kV voltage with a CuKα ray (λ = 1.5418Å). The data were treated with a Bragg Brentano Geometry processing system. All samples showed high crystallinity, as reflected by sharp XRD lines but different interlayer basal spacing according to the exchangeable cations (Appendix A). Additional measurements of the pH, Zeta potential and particle size were carried out at room temperature (RT) in 20 mL of pure water or aqueous suspensions of 40 mg clay powder in LA solution. This was achieved using Malvern-Zetasizer S90 Nanoseries equipment (operating with a 90-Plus particle sizer software, version 4.20) and a Brookhaven device (Zeta-Plus/Bl-PALS, Holtsville, NY, USA)

### 2.2. Calibration Curves and Ozonation Experiments

Lactic acid (L (+), analytical grade from Sigma Aldrich, Oakville, ON, Canada) was previously dissolved in nanopure water at various concentrations (1, 2.5, 3, 4 and 5 g L^−1^). The resulting solutions were used for plotting calibration curves for further UV-Vis spectrophotometry (UV-Vis) and high-performance liquid chromatography analysis coupled with UV detection (HPLC-UV). Previous adsorption tests were performed after dispersing 40 mg of clay mineral in 20 mL LA aqueous solution (2.5 g L^−1^ or 27.75 mM) under vigorous stirring for 3 min. Further, each sample was additionally stirred for different times, then centrifuged and filtered with a 0.22 µm filter. The supernatant was analyzed.

Ozonation attempts were carried out using various 20 mL LA aliquots of 2.5 g L^−1^ (27.75 mM) concentration in nanopure water at intrinsic pH in a cylindrical PVC vessel (28 mm × 115 mm) at RT. The ozone was generated produced from air by an A2Z-AQUA-6 portable ozone generator (A2Z Ozone Inc., Louisville, KY, USA) and bubbled in these samples for different times ranging from 20 seconds to 60 minutes at a constant 600 mg h^−1^ throughput. For catalytic ozonation, 40 mg of dry clay catalyst were added under vigorous stirring for 3 min to 20 mL LA aliquots. The hydrophobic character and hard delamination of Ht imposed longer stirring of 2 h before ozonation.

### 2.3. Reaction Mixture Analysis

The supernatants of the ozonized reaction mixtures were then centrifuged and analyzed at RT with two complementary methods. UV-Vis spectrophotometry was used for a qualitative assessment of the reaction evolution in time and detection of the chemical functions produced through ozonation by means of an Agilent-Cary 60 instrument (Agilent Technologies, Santa Clara, CA, USA) monitored by an accessory computer and 1 cm quartz cell. LA UV-Vis spectra at different concentrations gave highly linear calibration curves for all the three absorption bands (Appendix A). The molar absorption coefficients were 178.0, 79.3 and 42.2, respectively (Appendix A). These calibration curves allowed monitoring the intensity increase in clay suspension in non-ozonized LA solution (Appendix A) and the evolution in time of the intensity of the main UV-Vis bands during ozonation (Appendix A).

The most intense band (195–200 nm) corresponds to the π→π* transition. This band shades LA degradation, often overlapping with those of early oxidized derivatives in the ozonation of many organic substrates [36,37,38,39,41,42,54]. The lower intensity shoulder appear at 210–215 nm, corresponding to the n→σ* transition, a specific feature of the carbonyl group in an aqueous solution. The barely detectable band observed at 265–290 nm was assigned to an n→π* transition of LA carboxyl.

The rates of LA conversion and product distribution were quantitatively determined with high performance liquid chromatography with UV-detection (HPLC-UV). This was achieved by injecting 3 μL samples of the ozonation mixture into an Alliance instrument coupled with a UV detector (WATERS-2487 dual **λ** absorbance) and an Agilent Eclipse Plus C18 column (3.5 µm, 4.6 mm × 250 mm). The latter operated at a constant 1 mL.min^−1^ flow rate of the mobile phase (isocratic mod, 0.06 mL trifluoroacetic acid, TFA per 1 liter nanopure water). The data were processed by a STAR acquisition software. The 210 nm band was found to be suitable for unequivocally identifying LA at a 3.53 min retention time with a ±1% standard deviation and detecting most ozonation derivatives which commonly adsorb at this band (Appendix A). This procedure turned out to be fairly reproducible as compared to other methods [55,56], resulting in highly linear calibration curve with a 0.0366 slope and a 0.9986 correlation coefficient (Appendix A).

### 2.4. Intermediate Identification by LC-ToF-MS

The samples were diluted to a 1:1 volume ratio with HPLC grade water, and 0.5 µL aliquots were analyzed with LC-ToF-MS through a Gemini-NX C18 column (50 mm × 2.00 mm, 3μm particle size, Phenomenex, Torrance, CA, USA) maintained at 30 °C. This was achieved by means of a reversed phase liquid chromatograph (1260 Series HPLC System) and a time-of-flight mass spectrometer (6224 TOF mass spectrometer), both from Agilent Technologies (Santa Clara, CA, USA). The compounds were separated using 0.1% formic acid in ultrapure water (Solvent A) and methanol (B) under a 0.4 mL/min throughput and the following volume gradient: 0–2.0 min, 0% B; 2.0–9.5 min, 0–40% B; 9.5–10 min, 40% B; 10.0–10.5 min, 40–0% B; and 10.5–15 min, 0% B. The mass spectrometer was operated in negative electrospray mode under a 3.5 kV voltage (Fragmentor at 120 V) and a source temperature at 300 °C, with a dual spray configuration for internal calibration and high mass assessment accuracy. Mass spectra were acquired from *m/z* 40 to 1400 with an acquisition cycle of 0.89 s and a resolution higher than 10,000.

## 3. Results and Discussion

### 3.1. LA UV-Vis Spectrum Changes in Clay Suspension before Ozonation

Changes in the UV-Vis spectrum of the centrifuged supernatant of the non-ozonized LA solution previously contacted with the clay materials (Appendix A). Here, the potential LA–catalyst interactions, must involve π→π* transitions in the region 195–200 nm and n→σ* transition at 210–215 nm for the C=O and –C-O groups (Figure 1).

As compared to the ozonation of LA solution (*Pattern 1-curve in dashed line*), the addition of HMt-4, HMt-8 and HMt-24 induced a marked absorbance decrease (*Pattern 6, 7* and *9*), much less visible for HMt-15 (*Pattern 8*). This is a general tendency that can be explained by lactate anion depletion via protonation, LA dimerization and adsorption by non-electrostatic interaction as a result of lower pH (2.42–2.83) than LA pKa (3.86) induced in the LA solution by all cationic clay catalysts (Table 1).

Their higher Si/Al ratios are expected to confer them higher surface acidity as compared to the other clay minerals. This effect seems to be reversed or at least mitigated for bentonite, HMt1, NaMt and Ht (*Pattern 2–5*), which rather produced absorbance increases, presumably due to an enhancement of LA dissociation into dissolved lactate anion. This result is of great interest because it demonstrates that LA adsorbs via interactions depending on the catalyst behavior in aqueous media.

### 3.2. Effect of Clay-Induced pH on LA Adsorption

LA adsorption, if any, should involve clay–LA, LA–water and clay–water interactions with possible synergy and/or competitivity. LA adsorption on Ht after 3 min contact was reflected by lower LA residual amounts (LARA) of 54.4%, ca. 75.3% on KMt, 77.3% for both HMt-8 and HMt-15, 80.3% on bentonite, 87.1% on NaMt, 93.7% for HMt-4, 94.6% on HMt-24 and approximately 96.3% for HMt-1 (Figure 2). Almost no adsorption was registered on CaMt and MgMt, presumably due to a sandwiching effect of bivalent cations that favors clay aggregation via clay–clay interactions and reduces the accessible surface.

These different affinities towards LA can be correlated to the specific pH changes observed in the different clay suspensions in both water and LA solutions. Except for NaMt (pH 6.88), clay dispersion in distilled water resulted in pH decrease from the initial pH value of distilled water (6.55) to different values according to the clay mineral (Table 1). This must be due to specific acid–base properties of each clay mineral which induces specific clay–water interactions.

The slight pH increase induced by NaMt suspension in water should originate from clay cation exchange with consecutive formation of basic NaOH (NaMt + HOH → HMt + NaOH). Bentonite should act similarly but with various metal cations resulting in more or less basic metal hydroxides that maintain pH (6.25) close to that of water (6.55). The progressive pH decrease from 5.27 to 4.25 suggests a similar but progressively attenuated phenomenon for the acid-activated counterparts due to a decreasing cation exchange capacity with the activation duration [53].

In spite of its intrinsic surface basicity, hydrotalcite also induced a slight pH decrease in water from 6.55 to 6.27. This must be due to partial anion exchange of carbonate anions by hydroxyls present in water [29]. The released substance is assumed to readily protonate at pH levels below the pka_1_ value of H_2_CO_3_ (6.37). The more pronounced pH decay down to 4.64 in LA solution must arise from a higher carbonate production via lactate anion exchange in Ht. In spite of its highest particle size of 9.98 µm that accounts for lower dispersion and reduced adsorption surface, Ht displayed the highest affinity towards LA. This suggests an LA adsorption via an acid–base interaction promoted by the well-known surface basicity of hydrotalcite.

Hydrophobic interactions, if any, must be attenuated by higher pH (4.62) of LA solution in the presence of Ht as compared to LA pKa (3.86). It results that lactate adsorption via anion exchange must also be involved [29]. The released carbonate anion is assumed to readily protonate at pH levels below the pka_1_ value of H_2_CO_3_ (6.37). This process should be enhanced by the relatively high intrinsic basicity of Ht which induced relatively high initial pH in pure water (3.94). Here, the occurrence of more or less pronounced hydrophobic LA adsorption and LA:LA interactions with LA dimerization according to the pH must also be taken into account.

### 3.3. Effect of Catalyst Addition on LA UV-Vis Spectrum

Preliminary 3 min ozonation tests for different initial concentrations of LA in the absence of catalyst revealed marked changes in the UV-Vis spectrum (Figure 3). As a common feature, all ozonized LA solutions (dashed lines) displayed higher UV-Vis intensity as compared to their starting counterparts (solid lines). These bands are not due to dissolved ozone, since the latter readily disappears after a few seconds. These bands are rather due to the rise of early LA intermediates that absorb in the same region [37,38,39,41,42,50,54].

The mere rise of such intermediates during ozonation provides evidence of LA reactivity towards ozone. Clay-catalyzed ozonation induced more pronounced increase in the intensity of the UV-Vis bands (Appendix A). This indicates unequivocally the beneficial effect of the clay catalyst on LA conversion enhancement. Nonetheless, the conversion yield cannot be accurately assessed due to the appearance of new bands that shade the UV-Vis bands of LA. Quantitative HPLC-UV measurements of LA concentration during ozonation are essential requirements for this purpose.

### 3.4. LA Depletion during Ozonation and Product Identification

As expected, HPLC-UV measurements during ozonation revealed a simultaneous intensity increase for the intermediate signals with LA depletion in both the absence and presence of catalysts. This is well illustrated by superimposed HPLC-UV chromatograms of LA solution after different ozonation times in the presence of HMt-24 (Appendix A). A more pronounced LA depletion (solid lines) was registered in the presence of all clay catalysts as compared to non-catalytic ozonation (dashed line) (Figure 4).

As a common feature, all clay catalysts gave lower LARA values ranging between ca. 10% for Ht and 30% for CaMt as compared to ozone alone (36%) after 5 min ozonation. This accounts for LA conversion yields of 90, 70 and 64, respectively. This beneficial role of catalyst addition was maintained for longer ozonation time, but was marked attenuated beyond 5 min reaction.

Based on the LA residual amount, the degradation yield reached ca. 97% for NaMt, 93% for KMt and Ht, 92% for bentonite and HMt-4, 91% for HMt-15, 90% for HMt-8, HMt-24 and HMt-1, 87% for MgMt, 84% for CaMt and 85% in the absence of catalyst after 15 min of ozonation. LC-Tof-MS analysis of the reaction mixture after 1 min ozonation in the presence of NaMt revealed the presence of two major peaks attributed to residual LA and an intermediate at retention times of 11.9 and 4.3 minutes, respectively (Appendix A). Similar chromatograms were obtained will all reaction mixtures resulting from ozonation without and with each of the investigated catalysts.

The intense peak at an 11.9 min retention time was assigned to lactic acid with an *m/z* value of 89.0258–89.0266 amu (Appendix A). The peak appearing at a 4.3 min retention time even in the early stage of ozonation seems to correspond to a mixture of at least five compounds such as C_6_H_10_O_5_ (*m*/*z* = 157.02 amu), C_8_H_10_O (*m*/*z* = 201.05 amu, C_7_H_8_O_7_ (*m*/*z* = 203.02 amu), C_10_H_8_O_6_ (*m*/*z* = 223.022) and C_12_H_7_O_7_ (*m*/*z* = 263.02 amu). Among these, only that with a 161.0445 amu *m*/*z* appears to be close to C_6_H_10_O_5_ (162 g mol^−1^) denoted as intermediate 1 (Int-1).

This compound has many isomers, but lactyl-lactic acid and lactic anhydride are the most probable LA derivatives [57]. In this case, it appears that ozonation involves LA–LA and LA-derivative condensation processes that make the reaction mixture difficult to analyze. Investigation is still in progress in this direction.

The increase in time of the Int-1 peak (Figure 5) took place simultaneously with LA depletion, and was more pronounced during the first three minutes of ozonation in the presence of all clay catalysts, and even faster with Ht (Figure 4).

A marked attenuation was observed at 5 min ozonation, most likely due to competitive ozone consumption in Int-1 decomposition and other reactions. Among these, direct LA oxidation by molecular ozone generates acetaldehyde (CH_3_CHO) and carbon dioxide [34,35], which readily oxidize into acetic acid (CH_3_COOH) that may also react with residual LA molecules giving rise to Int-1 (C_6_H_10_O_5_). By analogy to animal metabolisms, LA oxidation should produce acetaldehyde and short-chain acetic, pyruvic and oxalic acids [58,59,60].

All these derivatives appear even in early and fast steps of ozonation, and their consecutive oxidation gives rise to a wide variety of intermediates, among which most still remain to be identified [61,62,63]. Two very low intensity peaks were also detected at a retention time around 2 minutes for ozonized LA solution in the presence of NaMt and were attributed to other LA derivatives. Both peaks totally disappeared after a few additional minutes and were not observed for some catalysts such as bentonite, HMt-24 and others.

### 3.5. Ht-Catalyzed Ozonation

As compared to smectite catalysts, Ht produced the fastest depletion of LA which accounts for ca. 88–89% conversion yield after only 2 min and 90% after 5 min ozonation (Figure 6). These values are ca. twice higher than LA retention yield on Ht, reflected by a 53–54% LARA after 3 min adsorption in the absence of ozone. It results that Ht display adsorptive and catalytic activity in LA ozonation with possible synergy of both properties, in agreement with the literature [64]. The residual amount of LA barely reached 38–40% when Ht was stirred manually for 5 min during ozonation but dramatically dropped down to ca. 10% after 5 min ozonation under vigorous stirring. Longer ozonation time of 15 min allowed affording LA conversion yield of up to ca. 75% under manual stirring and around 92–93% under vigorous stirring.

Hence, the contact time and strong convection turn out to be essential requirements for overcoming the hard LDH delamination and diffusion hindrance between clay lamellae [65]. This can be explained in terms of the low hydrophilic character that imposes slow LA diffusion towards the catalytic surface due to weak Ht dispersivity and structure compaction in aqueous media.

Interestingly, this beneficial effect of strong convection appears to be limited by the appearance of a plateau with an almost constant 90-93% LA conversion yield between 5 and 15 min ozonation. This must be due to a detrimental pH decrease during ozonation that is supposed to alter the stability of the colloidal Ht suspension [66]. This is known to promote clay coagulation–flocculation, accentuated by LA protonation and the rise of hydrophobic aggregation of Ht lamellae.

### 3.6. Effects of pH Evolution

Starting from specific initial pH of LA solution (Table 1), all clay catalysts produced a marked pH decrease after 2 min ozonation (Figure 7). This pH decrease was continuous and progressively attenuated in time for all smectite-based catalysts. This stage is supposed to involve fast LA decomposition into acidic species such as pyruvic (pKa 2.45), acetic (pKa 4.75), oxalic (pKa_1_ 1.25 and pKa_2_ 4.3), formic (pKa 3.74) and carbonic (pKa_1_ 6.35 and pKa_2_ 10.32) acids. As compared to smectite catalysts, Ht produced a more pronounced pH decrease, but followed by an apparently paradoxical pH increase after 2 min LA ozonation (Figure 7a).

The most plausible explanation resides in carbonate release via anion exchange by lactate that generates carbonic acid (pKa_1_ 6.35 and pKa_2_ 10.33) at the expense of LA (pKa 3.8). This phenomenon seems to be enhanced by LA ozonation and the rise of carboxylic acids whose anions may also substitute carbonate. In spite of this pH increase, the reaction mixture still remains acidic where direct ozonation molecular ozone should prevail. Unlike in radical driven ozonation [67] and in neutral to alkaline media [68], the initial pH of the LA solution varies according to the clay suspension. The pH should govern the interactions of all the species involved in the adsorption and ozonation processes [64].

The marked pH increase during the first 15 min of ozonation suggests a massive carbonate release that shades the formation of acidic derivatives which also oxidize into carbonic acid. However, this apparently high catalytic activity of Ht contrasts with its highest particle size of 9.98 µm in LA solution (Table 1) and the expectedly lowest accessible catalytic surface. This must be due to its lowest absolute value of the Zeta potential (+16.23 mV) which accounts for low charge density on Ht surface and weak inter-particle repulsion forces.

This suggests a progressive delamination of Ht as a result of pH change during ozonation and a rise of alkyl carboxylates that promote anion exchange. Possible contributions of LA adsorption via strong hydrophobic and Lewis acid–base (LAB) interactions with Ht surface should also be taken into account. This is in agreement with the key role of the acid–base property of MgAl-LDH, which is supposed to promote synergy between acidic adsorption sites for ozone and base sites for catalytic oxidation of the organic substrate [69].

Hydrophobic inter-particle interaction was also suspected for HMt-24, given its higher particle size (5.51 µm) as compared to the other Mt-based catalysts. This can be explained by its highest Si/Al mole ratio (4.36), as a result of advanced dealumination [53]. Such interactions are expected to be more pronounced on HMt samples and with decreasing pH below the pKa value of LA (3.8), and are essential requirements for improved catalytic activity.

Here, the extent of the accessible catalytic surface should play a key role since NaMt displayed the highest catalytic activity reflected by the lowest LARA of 3% after 15 min ozonation and the highest dispersion in the aqueous media with lowest particle size (0.63 µm) as compared to all catalysts (Table 1). Removal of dense silica phases through bentonite purification was found to increase the Zeta potential from −25.73 mV (Bentonite) to −37.07 mV (NaMt), thereby enhancing the interlamellar repulsion forces and improving the accessible surface for adsorption and catalysis. Undissociated LA molecules may interact with hydrophobic siloxy groups (-Si-O-Si-) and/or with both in-plane and out-of-plane silanols, depending on the pH of the reaction mixture [37,70].

### 3.7. Ozonation with Ht-cationic Clay Mixtures

The appreciable effectiveness of cationic clay minerals and hydrotalcite has stimulated research on their combination at different proportions. HPLC-UV measurements showed no clear improvement as compared to ozonation by individual catalyst after 15 min ozonation since Ht still displayed the lowest LARA (Figure 8). Nevertheless, deeper insights into longer ozonation times of up to 60 minutes after 3 min impregnation under manual stirring revealed lower LARA values ranging from 2.63 (20–80% Ht-HMt-15 mixture) to 3.21% (20–80% Ht-HMt-1 mixture) than for Ht alone (3.31%) (Table 2).

Even though these LARA values are higher than those produced by most smectite-based catalysts after 60 min ozonation, HT-HMt mixtures still remain more interesting because of the beneficial presence of Ht that has adsorptive and catalytic properties. Interestingly, the strong LARA discrepancy between HT alone and certain HMt mixtures appears to be progressively attenuated up to total disappearance after 15 min ozonation. This must be due to Ht delamination by intercalation with smectite lamellae that reduce surface hydrophobicity.

Surprisingly, the ozonized reaction mixtures resulted in clear supernatants through easy and fast centrifugation. This suggests an increase in the hydrophilic character that promotes delamination. This result is of great interest because it suggests that reciprocal Ht and HMt intercalation improves the accessible surface and catalytic activity, and that optimum Ht-HMt mixtures can promote synergy. Research is still in progress in this regard.

### 3.8. Ozonation Kinetics

HPLC-UV measurements of short ozonation times allowed approaching a preponderant LA ozonation reaction with minimal contribution of intermediate transformations. Plotting C_LA_, 1/C_LA_ and Ln (C_LA_/C_LAo_) versus the reaction time, where C_LA_/C_LAo_ is the relative LA concentration allowed applying the zero-, first- and second-order kinetics models (Appendix A). The closest values of the correlation coefficient (R^2^) to unity among those summarized in Appendix A–S4 allowed establishing the most probable reaction order and rate constant with their corresponding range of ozonation time for each clay catalyst (Table 3).

As a common feature, most clay-catalyzed ozonations seem to trigger and keep proceeding through second-order kinetics that depends on the concentrations of both LA and ozone. Kinetic changes into the first order were only observed after 3–5 min ozonation in the presence of Ht, MgMt and HMt-24 and were maintained during the entire ozonation time investigated once maximum ozone solubility is attained. This confirms once again the occurrence of specific interactions of the clay surface with both reagents in the aqueous media.

These kinetic changes registered after the first 5 min of ozonation must also be due to pH decrease and change in the surface interaction, as previously stated. Decrease in pH is expected to enhance lactate anion protonation into non-dissociated LA molecules, whose adsorption, if any occurs, should involve only hydrophobic interactions, preferably on hydrophobic surfaces such as that of HMt samples and hydrotalcite. Investigations are still in progress in this direction.

## 4. Conclusions

The results obtained herein allow concluding that clay minerals display different catalytic activity in the ozonation of lactic acid in aqueous media according to their surface interactions with both reagents. These interactions are strongly dependent on the pH of the reaction mixture, which determines the dissociation of lactic acid molecules, clay dispersion in the aqueous media, particle size and the extent of the accessible catalytic surface. The catalytic activity of bentonite, montmorillonite and acid-activated counterparts appears to involve mainly hydrophobic adsorption of lactic acid molecules at low pH and proton retention from lactic acid dissociation at higher pH. On hydrotalcite, the catalytic ozonation seems to proceed via hydrophobic interaction and anion exchange. Hydrotalcite–smectite combination promotes clay dispersion that improves the catalytic activity. This result is of great importance because it allows envisaging potential application of layered double hydroxides in lactic acid extraction from dairy fermentation broths with consecutive regeneration by ozone in aqueous smectite suspension. This opens promising prospects for using harmless and recyclable natural materials in reducing the cost related to seed production in dairy technologies.

## Figures and Tables

**Figure 1 molecules-27-06502-f001:**
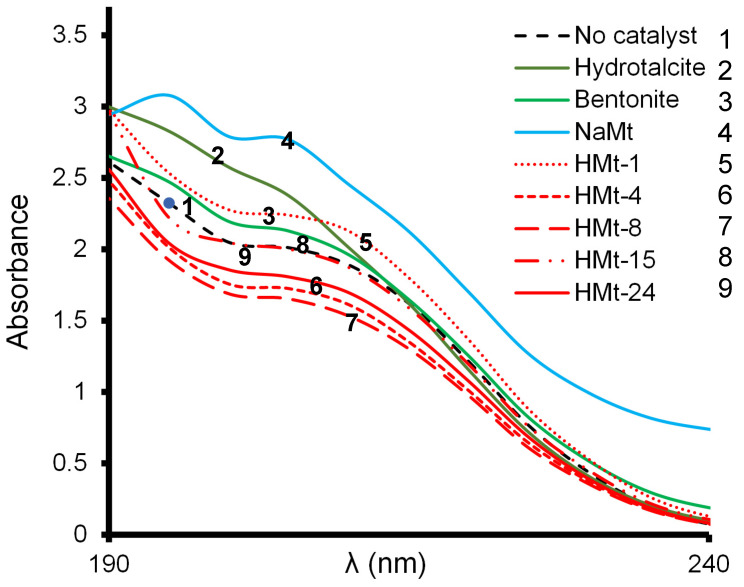
UV-Vis spectra close-ups of clay-free LA solution (dashed line of curve 1) and of the supernatant of the different clay suspensions in the 190–220 nm region. These spectra were recorded at T = 20 °C and intrinsic pH of the aqueous clay suspensions. Quartz cell = 1 cm. Initial concentration 2.5 g L^−1^ (27.75 mM). Contact time: 20 min.

**Figure 2 molecules-27-06502-f002:**
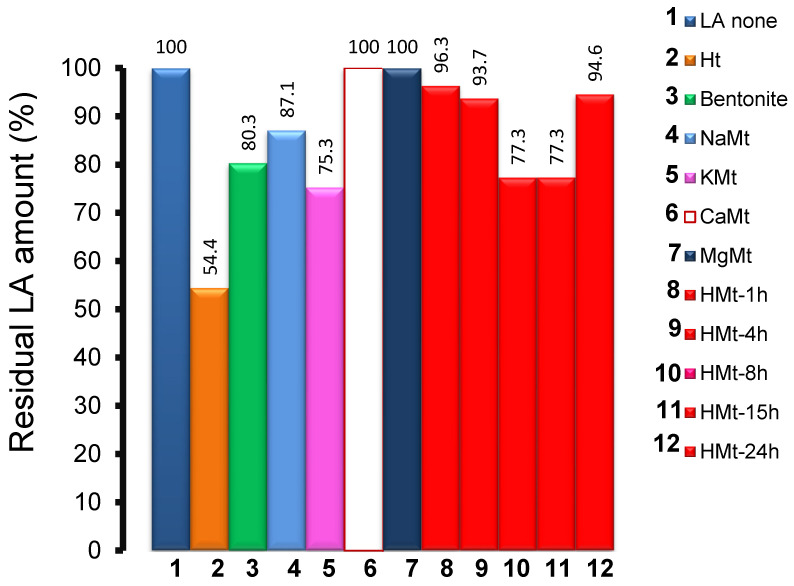
LA residual amount (%) in aqueous clay suspensions before ozonation as assessed by HPLC-UV in the centrifuged supernatant after 3 min contact with the respective clay mineral. Clay impregnation was achieved by mixing 4 mg of cationic clay minerals with 20 mL LA aqueous solution with a concentration of 2.5 g L^−1^ (27.75 mM) under manual stirring for 3 min. Ht was impregnated according to the same procedure but with 2 h stirring. The sample was then centrifuged and filtered with a 0.22 µm filter.

**Figure 3 molecules-27-06502-f003:**
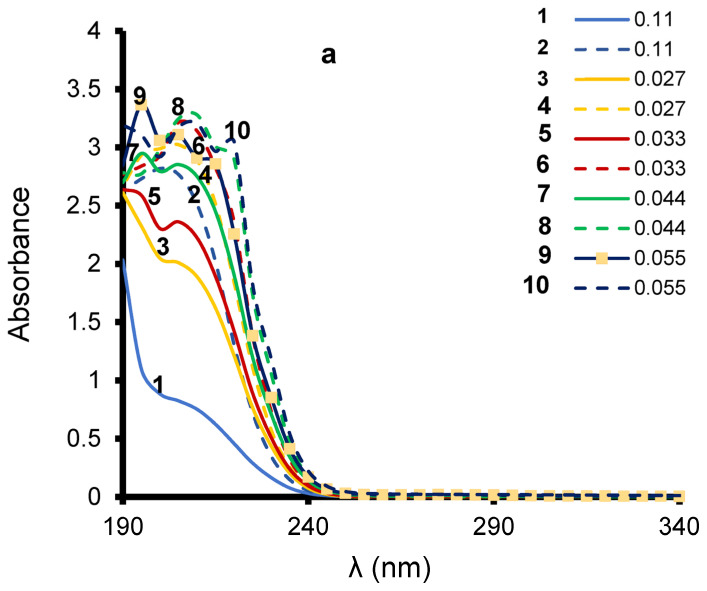
UV-Vis spectra of LA at different initial concentrations (mg L^−1^) before (solid lines) and after 3 min ozonation (dashed lines) in nanopure water (solid lines). T = 20 °C; pH = 2.80; quartz cell = 1 cm; ozone flow rate: 600 mg h^−1^. Prior to UV-Vis analysis, the sample was centrifuged and filtered with a 0.22 µm filter.

**Figure 4 molecules-27-06502-f004:**
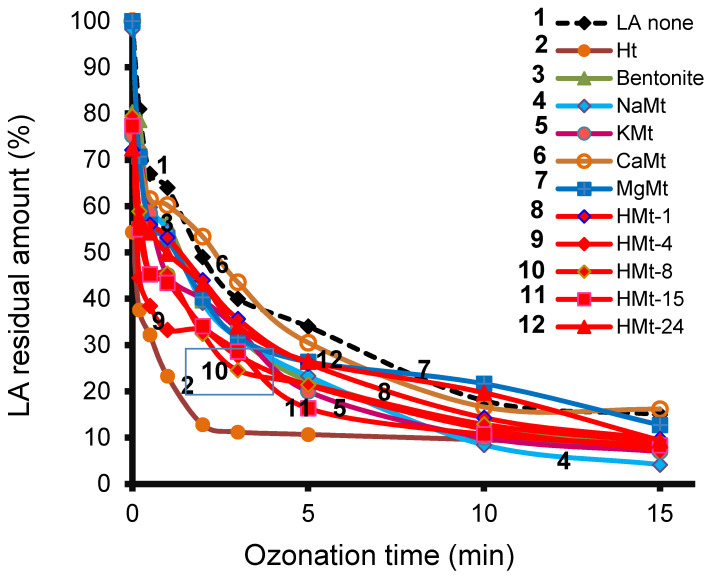
Effect of the addition of catalyst on LA residual amount during ozonation as assessed with HPLC-UV. Before ozonation, all cationic clay suspensions were stirred manually for 3 min for homogenization. Ht suspension was previously prone to a vigorous magnetic stirring for 120 min. After ozonation, the sample was centrifuged and filtered with a 0.22 µm filter.

**Figure 5 molecules-27-06502-f005:**
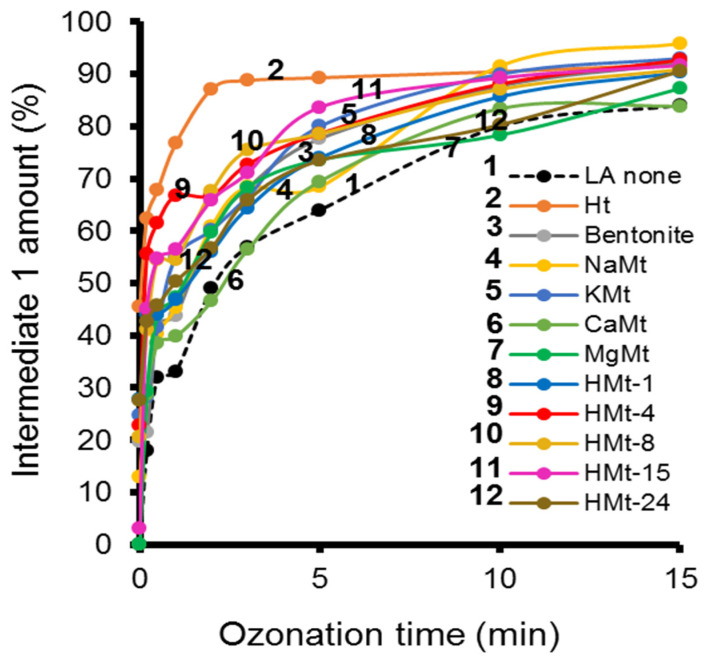
Evolution of the amount of intermediate-1 during ozonation in the presence of different catalysts as assessed by HPLC-UV. After ozonation, the sample was centrifuged and filtered with a 0.22 µm filter.

**Figure 6 molecules-27-06502-f006:**
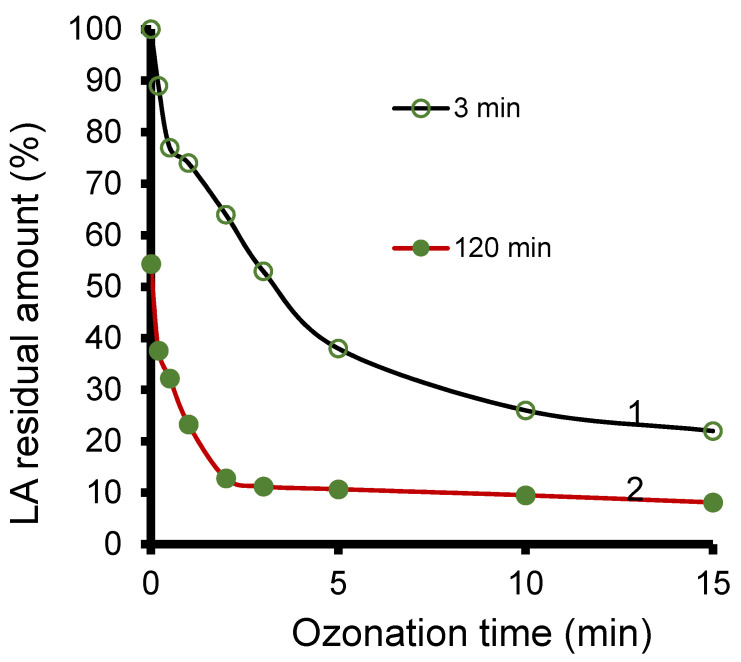
LA residual amount (%) during ozonation of Ht suspension after 3 min contact time under mild manual stirring (1) and 120 min under vigorous magnetic stirring (2). LA residual amount (LARA) was assessed with HPLC-UV in the centrifuged supernatant after 40 mg hydrotalcite immersion in 20 mL LA aqueous solution with a concentration of 2.5 g.L^−1^ (27.75 mM).

**Figure 7 molecules-27-06502-f007:**
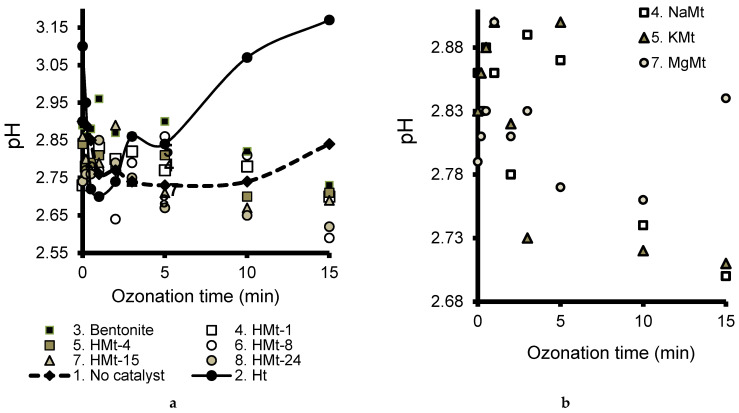
Evolution of the pH of the reaction mixture during LA ozonation in presence of various clay catalysts (**a**) and focus on ion-exchanged Mt samples (**b**) as compared to non-catalytic ozonation (*Curve 1-dashed line*) and hydrotalcite (*Curve 2*).

**Figure 8 molecules-27-06502-f008:**
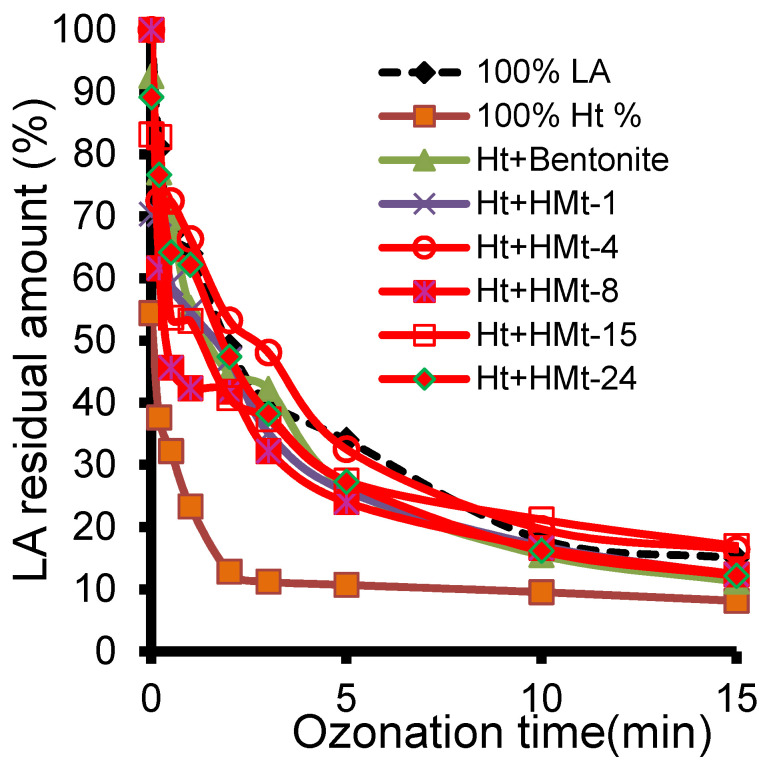
LA depletion in time during ozonation in the presence of 80–20% Ht-Mt mixture as determined by HPLC-UV. Since montmorillonite (Mt) is the preponderant clay mineral with expanded structure in bentonite and modified counterparts, the latter was denoted in this figure as Mt.

**Table 1 molecules-27-06502-t001:** Main behavior features of the investigated clay materials in water and LA solution.

Sample	Si/Al Mole Ratio	pH ^a^			Average Particle Size (µm) ^e^		Average Zeta Potential (mv)	
		Water ^b^	LA 0 min ^c^	LA 1 min ^d^	H_2_O	LA	H_2_O	LA
Bentonite	2.50	6.23	2.80	2.84	1.78	1.01	−30.87	−25.73
HMt-1	2.69	5.27	2.69	2.87	2.81	1.98	−18.72	−24.46
HMt-4	3.00	4.13	2.67	2.74	2.69	0.78	−24.93	−25.53
HMt-8	3.47	4.38	2.67	2.69	2.98	1.24	−27.10	−22.09
HMt-15	4.03	3.82	2.42	2.73	3.32	2.28	−35.93	−31.88
HMt-24	4.36	4.48	2.60	2.67	3.71	1.16	−21.48	−23.75
NaMt	2.45	6.88	2.76	2.82	0.63	0.63	−25.07	−37.07
MgMt	2.45	5.72	2.83	2.84	1.68	4.25	−15.85	−12.06
KMt	2.45	6.22	2.74	2.76	0.90	0.91	−18.91	−30.67
CaMt	2.45	6.07	2.74	2.74	1.27	1.12	−23.14	−19.62
Ht	Mg/Al=2	6.27	4.64	4.19	1.40	9.98	−09.33	+16.23

^a^ The initial pH values of distilled water and lactic acid solution were of 6.55 and 2.90, respectively. ^b^ Average initial pH of the clay suspension in water at room temperature. ^c^ Average pH of the aqueous LA solution containing the dispersed catalysts before ozonation. ^d^ Average pH of the aqueous LA solution containing the dispersed catalysts after 1 min ozonation. ^e^ Average particle size calculated from triplicate measurements.

**Table 2 molecules-27-06502-t002:** Comparison of the LA residual amount for ozonation with clay minerals and their mixtures.

Ht Content (%)	LARA (%) after 60 min Ozonation									
	Bent	HMt-1	HMt-4	HMt-8	HMt-15	HMt-24	NaMt	KMt	CaMt	MgMt
100	3.31	3.31	3.31	3.31	3.31	3.31	3.31	3.31	3.31	3.31
80	3.93	3.40	5.30	3.20	4.06	3.19	5.65	-	-	-
60	3.85	6.35	4.25	2.93	2.67	2.66	3.92	-	-	-
40	2	4.14	3.95	3.68	2.67	3.80	3.5	-	-	-
20	4.98	3.21	3.57	3.28	2.63	2.99	3.97	-	-	-
0	1.99	1.55	1.65	2.54	3.64	1.62	1.97	2.00	2.59	1.60

**Table 3 molecules-27-06502-t003:** Most probable reaction orders and rate constants for ozonation with different catalysts.

Catalyst	Time Range (min)	Reaction Order	R^2^	Rate Constant, K *
Ht	[0–3]	2	0.9613	1.039
	[5–15]	1	0.9999	0.136
Bentonite	[0–5]	2	0.9909	0.266
	[5–15]	2	0.9972	0.324
NaMt	[0–5]	2	0.9889	0.251
	[5–15]	2	0.9824	0.785
CaMt	[0–5]	2	0.9562	0.158
KMt	[0–5]	2	0.9672	0.288
	[5–15]	2	0.998	0.37
MgMt	[0–3]	2	0.9566	0.36
	[5–15]	1	0.9346	0.162
HMt-1	[0–5]	2	0.9448	0.195
	[5–15]	2	0.9997	0.26
HMt- 4	[5–15]	2	0.9905	0.36
HMt-8	[0–5]	2	0.9286	0.28
	[5–15]	2	0.9999	0.251
HMt-15	[0–5]	2	0.9513	0.355
	[5–15]	2	0.9971	0.234
HMt-24	[0–5]	2	0.941	0.192
	[5–15]	1	0.9428	0.274

* The rate constant (K) is expressed in g L^−1^ min^−1^ for the zero-order model, min^−1^ for the first-order model and L g^−1^ min^−1^ for the second-order model.

## Data Availability

Not applicable.

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
