# Peer review of "Clay-Catalyzed Ozonation of Hydrotalcite-Extracted Lactic Acid Potential Application for Preventing Milk Fermentation Inhibition"

_molecules, 2022, doi:10.3390/molecules27196502_

Round 1

Reviewer 1 Report

Research questions are well defined and within the aims and the scope of the journal. Material is accordingly defined. Methods are suitable, properly described and used in a way that is possible to replicate the experiments. The investigation is performed to good technical standards. It is no ethical problem involved. Conclusions are well stated and based on the results. Discussion is sound and relevant. The paper opens promising perspective for further harmless and recyclable application of natural materials in dairy technologies.

Suggestion:

Figure 2. At the vertical dimension it should be correct »Residual«.

Author Response

A point-by-point response to the reviewer’s comment is herein attached

Reviewer 2 Report

The manuscript entitled "Clay-catalyzed ozonation of hydrotalcite-extracted lactic acid Potential application for preventing milk fermentation inhibition" is an interesting area of research. Here authors have described/observed an unprecedented route for mitigating the inhibitory effect of lactic acid (LA) on milk fermentation through lactate adsorption on hydrotalcite (Ht) from simulated lactate extracts. The article may be very useful to readers globally. There are some minor mistakes observed in the form of language, grammar, comma and punctuations etc. Thus, I recommend revision before it can be accepted.

Author Response

The same point-by-point response to the reviewer’s comment is herein attached
